# The R Language: An Engine for Bioinformatics and Data Science

**DOI:** 10.3390/life12050648

**Published:** 2022-04-27

**Authors:** Federico M. Giorgi, Carmine Ceraolo, Daniele Mercatelli

**Affiliations:** 1Department of Pharmacy and Biotechnology, University of Bologna, 40126 Bologna, Italy; ceraoc98@zedat.fu-berlin.de (C.C.); daniele.mercatelli2@unibo.it (D.M.); 2Department of Mathematics and Computer Science, Freie Universität Berlin, 14195 Berlin, Germany

**Keywords:** R, statistics, bioinformatics, programming, CRAN, data science

## Abstract

The R programming language is approaching its 30th birthday, and in the last three decades it has achieved a prominent role in statistics, bioinformatics, and data science in general. It currently ranks among the top 10 most popular languages worldwide, and its community has produced tens of thousands of extensions and packages, with scopes ranging from machine learning to transcriptome data analysis. In this review, we provide an historical chronicle of how R became what it is today, describing all its current features and capabilities. We also illustrate the major tools of R, such as the current R editors and integrated development environments (IDEs), the R Shiny web server, the R methods for machine learning, and its relationship with other programming languages. We also discuss the role of R in science in general as a driver for reproducibility. Overall, we hope to provide both a complete snapshot of R today and a practical compendium of the major features and applications of this programming language.

## 1. Introduction

Since its birth in the mid-1990s, the R programming language has evolved into a fundamental computational tool for research in several fields, including statistics, biology, physics, mathematics, chemistry, economics, geology and medicine. One of the fundamental features of the R language is its relatively easy learning curve, supported by its in-built plotting functionality, which allows even a beginner to be able to produce meaningful graphs. The power of R has expanded from a simple implementation of statistical methods to a living galaxy of features, with applications that reach into modern artificial intelligence and web development. In the following paragraphs, we provide a brief history of R, followed by a description of the main R resources and repositories. We also provide a description of the R core idea of merging data processing, statistical analysis and graphical representations, with some graphical and written examples. We discuss the current role of R in web development, pipeline reproducibility and machine learning, ultimately providing a non-exhaustive selection of user-friendly resources to learn R or to stay updated with the novelties of the R world.

## 2. History of the R Programming Language

### 2.1. The S Language

The roots of the R programming language lays in the S language, and, more generally, in the work performed by John Chambers at the Bell Laboratories [1]. During the mid-1970s, having established some of the major milestones of modern software, including the UNIX operating system and the C and C++ programming languages, the Bell Labs focused their attention on statistical analysis [2]. Before S, they had developed an efficient Statistical Computing Subroutines (SCS) library for the compiled programming language Fortran, but soon realized that the frequent non-standard statistical and day-to-day analyses required a more interactive environment, and the possibility to explore the data being analyzed [2]. Thus, between 1975 and 1976 (Figure 1), the Bell Labs statistical research teams (including John Chambers, Richard Becker and Allan Wilks) developed the S language (S being an informal name indicating “Statistics”), with the expressed aim, stated verbatim by Chambers, “to turn ideas into software, quickly and faithfully” [3]. The S language increased in popularity, scope, and efficiency and, by 1988, it was considered a stable and fully fledged programming language and interactive environment for statistical analysis, with the publication of a reference book by Chambers, “The New S Language” [4]. Many of the core functions of the current R were already present in the 1988 S version, and the R documentation still cites “The New S Language” as the reference for base functions such as *mean* (to calculate the arithmetic mean of a vector of numbers) and *rnorm* (to generate random numbers drawn from a normal distribution).

### 2.2. The Birth of R

At the beginning of the 1990s, S had become very popular amongst statisticians, and several implementations were created. One of these was S-PLUS, which provided an intuitive graphical user interface (GUI) but was released under a commercial license. During this period, Ross Ihaka was a newly appointed Statistics Professor at the University of Auckland (New Zealand), where he met Robert Gentleman, a Professor from the University of Waterloo (Canada) on sabbatical [5]. Their eventful meeting started the development of a language of their own for statistical analysis, teaching and “trying out ideas” [1,5]. This new programming language was heavily inspired by the structure of the minimalist functional language Scheme, and by the scope, looks and syntax of both S, of which R can be considered an implementation, and S-PLUS [6]. Ihaka and Gentleman colloquially named the new language “R”, both as an alphabetical successor to “S”, and after their first name initials, and started advertising it on the S mailing list in 1993 [7]. Technically, most S code could (and still can) be run by an R interpreter, and vice versa, as R follows the S language definition from “The New S Language” 1988 book as much as possible. A few differences were introduced early on; for example, R adopted *lexical scoping*, allowing variables to be called within a specific block of code without being defined globally. Moreover, minor divergences were included to improve the “quality of life” of the R programmer; for example, the code list1[i] <- NULL will result in the removal of the element i from the list “list1” in R, but not in S, where it returns an error. Another example lies in the variables “T” and “F”, which are equivalent to Boolean TRUE and FALSE in both languages; but while “T” and “F” are reserved words in S, they can be overwritten in R, allowing for the creation of variables from all letters of the alphabet [8].

As R gained popularity in the years 1993–1994, many academical users started offering their voluntary contribution to the project. Ihaka and Gentleman were convinced by the newborn community (in particular by Martin Maechler of ETH Zurich) to release the whole language as free software, and they released the R source code under the GNU general public license in the June of 1995 [7]. This ensures, until the present time, that the R core software remains free to use and is not exploitable for commercial uses. In retrospect, the decision of making R freely available as a tool for the whole of mankind contributed to its immense success in the following years, and the R language overtook S and any other S derivation [9].

### 2.3. The R Community

After its release as free software, R emerged rapidly as an accessible and powerful tool for data management, analysis and visualization. This was made manifest already in the title of the first international peer-reviewed R paper: “R: A Language for Data Analysis and Graphics” [1], authored by Ihaka and Gentleman in 1996 in the Journal of Computational and Graphical Statistics. The development of the Internet allowed for the quick generation of an international community of R users, statisticians and non-statisticians alike. In 1997 this inevitably led to the generation of the first online aggregation space for R, in the form of three mailing lists: r-announce, r-devel and r-help, hosted by ETH in Zurich [10]. The official mailing lists, especially r-devel and r-help, are still used to the present day, despite the fact that the mailing list format has been overtaken by newer online aggregators able to rank questions and answers and popularity, such as the programming-oriented Stack Overflow or the statistics-oriented Cross Validated. The size of the messages archived from April 1997 to the present day are very informative of the initial boom of R popularity (years 2001–2013), followed by a decline in the mailing list as the prime instrument for community aggregation on the Internet (Appendix A).

The growing R community started writing dedicated expansions to the R original code, called libraries, and required an official packing system and software repository. This led to the creation of CRAN, the Comprehensive R Archive Network, which was announced in the r-announce mailing list by Kurt Hornik in 1997 [11].

In June 1997, the R community gave itself an official leadership in the form of the R core team, constituted by Ross Ihaka, Robert Gentleman, John Chambers (the original author of S), Martin Maechler and other “wise people” bestowed with write access to the R source code [12]. As a final step to building solid foundations for R, the core team launched the official R website (https://r-project.org, accessed on 21 April 2022) in 1999 [13].

### 2.4. The Official R Project after the Year 2000

R reached its first stable release version (1.0.0) on 29 February 2000 [14]. As R and its community increased in size and scope, so did the requirements for maintaining the CRAN server mirrors and supporting the core developers; thus, a non-profit organization was created to support it, the R Foundation, in 2003 [15]. The R Foundation statute is active to this day to promote and administer the R project and to act as a reference point for the R community [16].

R has frequent update releases, and it has, at the time of writing, reached version 4.1.3. Major updates (2.0.0, 3.0.0 and 4.0.0) did not occur as radical changes to R, but rather to testify that R had reached a consolidated maturity over the previous major version. For example, version 2.0.0, released in 2004, introduced the *lazy loading* functionality, which allowed large files to be loaded on the R environment only in the portions needed by the user, without using excessive memory. This R improvement allowed the language to be ready for the data explosion that occurred in the subsequent years, fueled, e.g., by novel technology for quantitative measurement; one example was the emergence of microarrays and next-generation sequencing in the field of biology [17]. R 2.0.0 therefore crowned the efforts undertaken in the years 2000–2004 to address the limitations in handling big data [5]. R, across the years, has always been aware of the outside world, both in terms of scientific requirements (e.g., the larger datasets of the 21st century), and in terms of technical possibilities offered by improvement in the operating systems and hardware on which R was running. Hence, R 3.0.0, released in April 2013 [18], completed the previous decade’s effort in making R able to fully utilize 64-bit architectures; this allowed it to handle numbers and indexes larger than 2^31^, and to manage object sizes having a memory size above 4 GB (and up to 8 TB). However, up to version 4.1 (the most recent at the time of writing, April 2022), R can be compiled in both 32-bit and 64-bit systems, and its release for Windows provides users with builds for both architectures, in order to allow even older machines to be able to run the most recent R code (https://cran.r-project.org/bin/windows/base/, accessed on 21 April 2022).

Seven years after R 3.0.0, and more than twenty years after R 1.0.0, R 4.0.0 was released in April 2020 [19]. It introduced several further optimizations to memory allocation (e.g., when an object is copied into another object) and “quality of life” improvements, especially meant for new users. One much welcomed improvement of R version 4 is that string data are now imported as character objects: an early developed memory-saving default parameter of R had been, up to this point, to import string data as factors, which had advantages but was counter-intuitive for beginners.

### 2.5. The Founding of Bioconductor

The creators of R and the R core team, while monitoring the proper development of the R language, have historically always welcomed external contributions, expansions and re-imaginations of the original project. The CRAN repository was created in 1997 as an environment for general R code and packages, but it soon became apparent that a specialized project for bioinformatics R users was needed. The early 2000s witnessed the beginning of an avalanche of biological data being generated, e.g., after the broad diffusion of microarrays [20] and the release of the first higher eukaryote genome sequences [21]. Thus, under the supervision of Robert Gentleman, Bioconductor started in 2001 [22], with the broad goal of creating and developing R tools for the analysis of biological data, usable by both statisticians and computational biologists. From the outset, Bioconductor received institutional funding and support, and grew quickly, seeing its first stable release (1.0.0) in 2002. The Bioconductor project now collects thousands of packages and is the de facto main aggregator of computational tools to analyze biological data, especially quantitative omics data (more on Bioconductor in the following paragraphs) [23].

### 2.6. Expanding R: ggplot2 and the Tidyverse

On 10 June 2007, the programmer and statistician Hadley Wickham, then a PhD student at Iowa State University, developed a graphical R package that would become not only more used than the base R plotting functions, but start a grammatical revolution in the R world [24]. *ggplot2*, in its simplest form, is an R package (downloadable from CRAN) that adds convenient functions to graphically explore data, and the elegant “look and feel” of its graphs has become quite popular in the R community [25].

*ggplot2* can now be considered to be a strong alternative to the R core plotting functions, based on the core packages *lattice* and *grid*. Despite differences in the data input format and operations, both *ggplot2* and base R can achieve identical graphical results, especially since the R 4.0.0 update, which introduced a more aesthetically pleasing standard color palette (Figure 2A).

*ggplot2*, together with other packages such as *dplyr* for data manipulation and *tibble* for data storage, also authored by Wickham, were united to form the *tidyverse* on 15 September 2016, a set of packages that collectively re-imagine the data flow operations in R, introducing the UNIX concept of piping with the form “% > %” [26].

### 2.7. Beyond Statistics: RStudio, Shiny and Rmarkdown

The tidyverse, including popular packages like *ggplot2* and *dplyr*, is currently maintained and promoted by the RStudio company (RStudio, Boston, MA, USA, https://www.rstudio.com/products/rpackages/, accessed on 21 April 2022). The eponymous tool of this foundation, RStudio, was the first IDE dedicated to R, and its work started in 2010 [27], with the first beta release appearing in 2011 [28]. RStudio was conceived and imagined not only as an editor to write and execute R code, but also as a growing universe for R development, and to extend the scope of the programming language beyond statistical analysis. Two milestone packages, in particular, were created by the RStudio team to pursue this objective. The first is *shiny*, launched in November 2012 [29], which extends the R functionality towards the web, allowing the language to produce stable and contemporary web resources [30]. The second is *rmarkdown*, stably released in 2014 [31], whose general scope is to provide reproducible pipelines and communicate results, via embedding R code and results in dynamic documents [32]. Specific R tools, expansions and the current state-of-the-art of the R world are covered in the following paragraphs.

## 3. The R Repositories

The R interpreter and console are provided as source code or executables for major operating systems from the CRAN website at https://cran.r-project.org/ (accessed on 21 April 2022). The basic R contains functions to perform all major statistical tests, plotting, and matrix operations, and, as of version 4.1.3, it is provided as a combination of 14 different core packages: *base*, *compiler*, *datasets*, *grDevices*, *graphics*, *grid*, *methods*, *parallel*, *splines*, *stats*, *stats4*, *tcltk*, *tools* and *utils* [8].

The 14 core packages, of course, do not cover the large world of R, which now spans from bioinformatics to web development. Thus, users have the possibility to install and load additional packages, or libraries, both from custom and official sources. Currently, there are three major repositories for additional R packages: CRAN, Bioconductor and R-Forge. These three require minimum standards of quality and active maintenance, and perform rigorous testing for all packages worthy of inclusion. However, many useful packages exist outside the three main resources, often in general code repositories such as SourceForge and GitHub. Once installed, every package can be then loaded with the library() command, which will also load all dependencies.

### 3.1. CRAN

At the date of writing (21 April 2022), the CRAN repository hosts **19,001** packages (source: https://cran.r-project.org/web/packages/), covering a large scope of applications. These packages extend the statistical capabilities of R, in addition to implementing novel graphical and technical methods, providing R with extended capabilities, e.g., for high-performance and parallel computing [33], in addition to the aforementioned shiny, tidyverse and Rmarkdown extensions, amongst many others. The process of having a package submitted, screened and ultimately accepted by CRAN may require several months for a beginner and requires the user to be fully aware of the CRAN repository policy (available at https://cran.r-project.org/web/packages/policies.html). Ultimately, packages in CRAN have the encouraging tendency to be well written and documented, providing cutting-edge and efficient methods for contemporary statistical analysis. Prior to submission to CRAN, an author should locally test its package with the following command, which automatically detects code inconsistencies, both fundamental and in terms of style:



check(args = c(’--as-cran’))



Because CRAN is intrinsically tied with R, installing packages from this repository has the easiest form of installation, via the install.packages() function. For example, to install the *GeneNet* package to infer gene coexpression via partial correlation [34], the user should simply type:



install.packages("GeneNet")



This function will also install dependencies, in binary format for operating systems such as Windows or MacOSX, or by compiling them in Linux systems. CRAN is also excellently integrated with RStudio, which can check for missing libraries and have the user install them with a simple click.

Although the install.packages() function is, by default, set up to search and work on CRAN only, it is possible to set it to install packages from the other two repositories. For example, the function setRepositories() will allow the user to set further locations for the installation process to look for. Currently (R version 4.1.3), packages from all three major repositories (CRAN, Bioconductor and R-Forge) can be installed this way.

One of the peculiar functionalities of CRAN is its package checking system, developed originally by Kurt Hornik [35], which implements simple text-based metadata and a clear hierarchical system, which has allowed, for decades, the maintenance of a healthy and consistent repository. CRAN packages are subject to continuous testing on multiple platforms and operating systems.

### 3.2. Bioconductor

Bioconductor is the second largest R package repository, hosting at this date (21 April 2022) **3422** packages (2083 software packages and 1339 data packages) (source: https://www.bioconductor.org/packages/release/BiocViews.html). Unlike CRAN, Bioconductor has a specific package focus, which revolves around bioinformatics and, in general methods, tools and data associated with biological studies. The process to have one package accepted by Bioconductor can be even longer than that of CRAN, following even stricter rules (including a maximum line width of 80 characters). The following function performs automatic checks on a package for Bioconductor rule compliance:



BiocCheck()



Bioconductor contains extremely useful tools for dealing with biological data, from differential gene expression analysis (e.g., *DESeq2* [36] and *limma* [37]) to genome analysis (e.g., *GenomicRanges* [38]). The Bioconductor code requires new packages to take advantage of the existing object classes, to ensure that authors do not have to constantly reinvent the wheel to represent, e.g., a transcriptomics dataset or a genomic region.

Bioconductor hosts some of most downloaded bioinformatics tools in the world (source: https://bioconductor.org/packages/stats/, accessed on 21 April 2022), but not all biocomputational R packages are hosted here; examples are *corto* for gene network reverse-engineering [39] and *Seurat* for single cell RNA-Seq analysis [40], which are hosted in the more general CRAN.

Bioconductor packages are not released in a continuous way, like those of CRAN, but in periodic versions of Bioconductor itself, which follow the release calendar of the main R full releases.

### 3.3. R-Forge

R-Forge, which hosted at the time of writing (21 April 2022) **2146** packages [41], is a collaborative R repository focused on developing packages, and providing additional tools for bug tracking, versioning and branching. It contains several unpublished prototype R packages, in addition to pre-release versions of CRAN libraries. The role of R-Forge is, as the name implies, to create and test novelties in R, with the help of a vibrant user community and before the strict requirements of the other two official repositories.

### 3.4. Github

Recently, part of the immense work performed in developing R code has moved onto the general environment of GitHub, arguably the most popular versioning system and developing location on the Internet. At the date of writing (21 April 2022), GitHub contains **34,268** active R projects [42], in several states of maturity. Although they may be frowned upon by the core R community, many scientific tools in R have been published without being on CRAN or Bioconductor [43,44]. Packages available on GitHub can only be downloaded and freely explored, but more recently a CRAN package, *devtools*, provided a convenient function to install them directly from the GitHub repository. The code to install the *svpluscnv* package for analyzing somatic copy number alteration events in cancer is as follows [43]:



library(devtools)





install_github("gonzolgarcia/svpluscnv")



GitHub (and other unofficial online repositories for R code, such as SourceForge) allow authors to quickly share their code without the stricter coding rules of CRAN and Bioconductor, but does not guarantee the validity, usability, execution and long-term maintenance of any of the code it provides.

## 4. Practical R

The practicality of R makes it an ideal first programming language to learn, since it is possible for beginners to obtain an immediate visualization of their own data following simple steps. Conceptually, the functionality of R can be divided into three classes, which taken together can summarize its role in bioinformatics, statistics, and data science in general: data interaction, analysis, and results visualization.

The following paragraphs provide some basic functions to visualize the core potential of R. The code can be directly executed on a standard R console (version 4.1.3 at the time of writing), available at the official R website https://cran.r-project.org/.

The following subparagraphs rely on small artificial datasets that are available as Appendix A.

### 4.1. Data Interaction

The main goal of R is to extract useful information from data; therefore, in many R pipelines, the first key step is loading the data itself onto the R environment.

R can import data from many types of sources: text files, binary files (such as Microsoft spreadsheets), web links or dedicated R data formats. Dealing with text files, R can load directly from classic character-delimited files into a common R matrix object.

For instance, the following code will read data from a comma-separated value (CSV) text file:



example<-read.csv("example.csv")



R can also load from text files with any type of delimiting character with the core function read.delim(). R can also import data from the common Microsoft Excel formats (xlsx and xls), in this case requiring an additional CRAN package, *xlsx*, and specifying the sheet number from which to read:



library(xlsx)





example<-read.xlsx("example.xlsx",sheetIndex=1)



Apart from the Apache *parquet* file storage format [45], R-specific efficient alternatives to save and read data are provided by the native file formats RDA and RDS. The RDA format can be used to store (save) multiple objects and load them in memory with the save object names:



save(a,b,file="ab.rda")





load("ab.rda")



The RDS file format uses the same archiving algorithm (reducing the effective size of the data on disk) as that of RDA, but it is commonly used to store an individual object, and then, upon loading, assign it to a specific object name, e.g.,



saveRDS(example,file="example.rds")





example<-readRDS("example.rds")



R provides convenient functions to quickly visualize the content of any object; for example, the dim() function provides the size of the object:



dim(example)



In our case, this is a matrix with 1000 rows and 3 columns.

Another convenient function is head(), which visualizes the beginning of any object. In our case, R visualizes that the columns correspond to three variables, g1, g2, g3, representing artificially generated gene expression values across 1000 samples (rows).



head(example)




       g1    g2    g3



Sample_1 5.425938 6.827846 0.7478255



Sample_2 6.152623 7.804399 2.1705011



Sample_3 4.990498 6.589876 2.7549418



Sample_4 5.801309 8.791750 6.2522507



Sample_5 4.658917 5.013827 9.6978252



Sample_6 4.833635 5.777115 5.5270391


Both RDA and RDS formats are effective for fast transfer of files between collaborating R users and as conveniently accessible storages for R data. As an example, we show how a fairly large numeric matrix (with 200 rows and 50,000 columns) can be saved in RDA or RDS formats, reducing by an order of magnitude both the disk size and time required for input/output operations (Appendix A).

The choice of RDS or RDA format is ultimately a matter of user taste and context appropriateness. Technically, RDA can be advantageous to save multiple objects, and, in fact, saving two objects within a single RDA object saves a few bytes over saving them as two separate RDS files.

In this particular case, individual columns (corresponding to artificial genes g1, g2 and g3) can be extracted and saved in three different objects for further analysis:



g1<-example[,"g1"]





g2<-example[,"g2"]





g3<-example[,"g3"]



### 4.2. Analysis

Even at basic level, R contains dozens of functions to perform statistical analysis to explore and extract information from data. The function summary(), for example, can provide a general overview of a numeric distribution, including minimum and maximum values, and the mean:



summary(g1)




   Min. 1st Qu.  Median    Mean 3rd Qu.    Max.



  1.221   4.349   4.996   5.016   5.682   8.756


In R, it is also possible to perform statistical tests with core functions; for example, the Shapiro–Wilk test for normality [46]. In this case, the test deems the first two distributions (g1 and g2) to be likely normal, and the third one (g3) not to be so.



shapiro.test(g1)




p-value = 0.1994




shapiro.test(g2)




p-value = 0.6653




shapiro.test(g3)




p-value < 2.2e-16


A Student’s *t*-test can be performed to check if two normally distributed vectors of observations, such as g1 and g2, are drawn from different distributions [47].



t.test(g1,g2)




 Welch Two Sample t-test



data:  g1 and g2



t = -18.315, df = 1805.5, p-value < 2.2e-16



alternative hypothesis: true difference in means is not equal to 0



95 percent confidence interval:



 -1.1007649 -0.8878124



sample estimates:



mean of x mean of y 



 5.016392  6.010681


The t-statistic result is shown, and the associated *p*-value, from which we can numerically infer that the two distributions are statistically different.

Another example of core R functionality is constituted by correlation tests. All three major correlation coefficients (Pearson, Spearman and Kendall) can be calculated [48], with the simple cor() function (which by default, with no extra argument, calculates the Pearson’s coefficient):



cor(g1,g2)




0.7037503


Alternatively, the cor.test() function can be used, which also calculates a *p*-value associated with the correlation coefficient:



cor.test(g1,g2,method="spearman")




 Spearman’s rank correlation rho



data:  g1 and g2



S = 53462464, p-value < 2.2e-16



alternative hypothesis: true rho is not equal to 0



sample estimates:



      rho



0.6792249


One core philosophy guiding R and its community is to provide a function for every existing statistical/mathematical method. It is very likely that a textbook test is already present within R, and the user can search for it using the ?? function, e.g., ??wilcoxon will provide the implemented methods for Wilcoxon Rank Sum Tests [49].

### 4.3. Visualization

Powerful, text-based results are harder to interpret than the visualization of the data itself, or the test being performed; for example, although the Shapiro–Wilk test provides a mathematical assessment of the normality of a distribution, it is always better to visualize the distribution itself. The following code allows the user to show the density distribution of g1, paired with the result of the Shapiro–Wilk test:



plot(density(g1),lwd=2)





p<-signif(shapiro.test(g1)$p.value,3)





mtext(paste0("Shapiro test p-value = ",p))



The output of this code concept is shown in Figure 2B. By combining statistical methods and graphical visualizations, R can quickly show, in this example, that the first two distributions are normal-like, whereas the third distribution is not. The intrinsic power and success element of R is to provide this combination at all levels, from these simple tests to highly complex calculations and datasets, such as UMAP performed in single cell analysis [50].

In a similar fashion, the difference between distributions g1 and g2 can be assessed via combining graphical representations of the two with boxplots, and *t*-test results (Figure 2C):



boxplot(g1,g2,names=c("Gene 1","Gene 2")





p<-signif(t.test(g1,g2)$p.value,3)





mtext(paste0("T-test p-value = ",p))



Another example of combining analytical and graphical statistics with R is provided by the correlation tests. The following lines of code generate a scatter plot with the function plot() and overlay the results of the correlation test (Figure 2D). The plot() function itself can be enriched with several parameters, such as pch to control the point shape, or col to control the color of the points:



plot(g1,g2,pch=20,col="cornflowerblue")





cor<-signif(cor(g1,g2),3)





mtext(paste0("Pearson’s Correlation Coefficient = ",cor))



Finally, one of the exceptional features of R is that it can overlap not only text (in the previous examples, the outputs of statistical tests) with graphics, but also several graphical layers. In the following example, the three distributions are shown using three tracks: standard box plots [51], beeswarm plots [52] and violin plots [53] in order to fully assess the features of the distributions. The beeswarm() and violin() functions require the homonymous CRAN packages. This particular instance of overlapping box plots, beeswarm plots and violin plots is colloquially referred to as “BBV plot” (Figure 2E).



library(vioplot)





library(beeswarm)





boxplot(g1,g2,g3)





beeswarm(list(g1,g2,g3),add=TRUE)





vioplot(list(g1,g2,g3),add=TRUE)



## 5. Rmarkdown and the Role of R in Scientific Reproducibility

Markup languages, such as HTML and LaTeX, allow the text content to be enriched by extra elements, such as color, font type and emphasis such as italics and underline. An original implementation of the idea of this literate programming for R, still in use, is the Sweave system [54]. Today, there exists a specific markup system for R, called Rmarkdown, which in its simplest form allows merging of descriptive text, R code and graphics/tabular output into an organized document. This is achieved by a specific formatting language that allows “chunks” of R code (or even that of other languages) to be included within formatted text, and used to produce several output types, such as documents, books, presentations and web pages, in PDF, HTML or Microsoft Word/LibreOffice/OpenOffice formats [31].

The Rmarkdown system was first introduced by the *knitr* package in 2012 [55] and is now fully supported by the *rmarkdown* package, whose purpose is to “to weave together narrative text and code to produce elegantly formatted output” [31].

The practicality of Rmarkdown can be witnessed by using RStudio, which fully supports the markup language and allows for the generation of preconfigured Rmarkdown files, simply by pressing “New File” and configuring the basic features, such as author name, date, document title and default output format. The generated Rmarkdown file is a text file containing both format blocks and code blocks (Figure 3A), and is commonly associated with the Rmd extension. The following header is written in Rmarkdown language and instructs the *knitr*/*rmarkdown* frameworks to start compiling an HTML document.


---



title: "Rmarkdown example"



author: "Federico M. Giorgi"



date: "12/6/2021"



output: html_document



---


Following the header, text blocks and code blocks can be intermixed. The following is a simple text including a level 2 header (##) following normal text with optional formatting (the single asterisk transforms the text within in italics).


## Example of Text Block



Descriptive text with *optional* formatting 


Complete guides on this language and syntax are available on the official Rmarkdown website, hosted by RStudio (https://rmarkdown.rstudio.com/, accessed on 21 April 2022).

Code chunks can be included using triple back quotes (‘) specifying the language (by default r) and including the code itself. The code will run during the generation of the document, and, if present, graphics will be included in the output. The following example shows a comment line and the command to graphically print 1000 values randomly generated from a normal distribution:



‘‘‘{r}





# Example of code block





plot(rnorm(1000))





‘‘‘



Rmarkdown is widely used to produce reports based on standardized pipelines, which would be repetitive if rewritten manually each time, both in the code parts and in the visualization of the results. A common example is the analysis of RNA sequencing experiments that share similar quality checks, data visualizations, differential expression analysis, pathway enrichment analysis, et cetera. Rmarkdown allows the creation of ready-made presentations for each new dataset, with minimal effort often limited to changing the data input parts and the supervision of the final results.

This efficient and diffused system coincides with the ever-growing call for data sharing reproducibility in data science and computational biology [56]. Rmarkdown is a system that allows authors to share not only raw data, but also the fully reproducible pipelines that transformed the uninterpretable complexity of large datasets into processed tables, reports and figures. Sharing the code and the data is an excellent academic practice, necessary for scientific transparency and for speeding up the sharing of knowledge. The existence of Rmarkdown allows scientists to share their work beyond the interpreted results and in the most usable form.

## 6. Writing R: Editors and Environments

Beyond the simplest R built-in console, R users can benefit from text editors, IDEs and graphical user interfaces (GUIs) to implement, execute, test and deploy their work in only one place, without the need for any extra effort. R IDEs in particular are designed to help and assist the R programmer by showing, in the same window, the code being written, the active R console/interpreter, and other visual interfaces such as, for example, the variable environment (showing which variables and libraries are loaded in memory), the graphical plot outputs (Figure 3B) and other elements showing the file system, the help pages, the history of previously typed commands, et cetera.

A variety of IDEs and text editors are available to use as an R developer (Table 1). Some are meant specifically for R or a set of languages, whereas others are language-agnostic, supporting multiple programming languages through plugins or extensions. The choice of the best virtual place in which to create R code, whether a text editor or IDE environment, is a never-ending debate and relies on the editor or IDE functionalities, the field of application, and the user’s personal requirements and preferences.

### 6.1. IDEs/GUIs

Since graphics is one of the main pillars of R [57], IDEs are naturally well received and appreciated amongst R programmers, because they allow, in the same window, the combination of both the code and the generated graphical output. Below, we list a selection of the most used R IDEs at the time of writing (April 2022).

#### 6.1.1. RStudio

RStudio is one of the most popular R development environments, especially amongst younger programmers [58]. Launched on 2011-02-28, RStudio started as an R-specific open-source IDE, written predominantly in C++ and Java (https://github.com/rstudio/rstudio, accessed on 21 April 2022). Subsequently, it has evolved into a bilingual environment, focusing on R and Python. The application is available as desktop (RStudio Desktop) or Linux server-based (RStudio Server) versions; for both, free and commercial editions exist. It runs on all major platforms, with the same interface. It combines a console, a syntax-highlighting editor with the tab-completion feature, graphics, history and help, into a single workbench. It supports reproducible research and literate programming by mixing code and text documentations. Accordingly, RStudio natively supports an interface with Rmarkdown (see previous paragraph), so that results can be rendered and communicated into HTML, PDF or Word reports. Moreover, RStudio supports project sharing by interfacing with GitHub and other versioning systems.

#### 6.1.2. Jupyter Notebook

Jupyter Notebook, an acronym of Julia, Python and R, is currently among the most popular development environments used by data scientists, due to its flexibility. It is a server–client application, launched in 2014 within the IPython project (https://ipython.org, accessed on 21 April 2022), and combines three components: the notebook application, kernels and notebook documents.

A Jupyter Notebook is saved as a JSON (JavaScript Object Notation) file (.ipynb), which makes it platform- and language-independent, and thus easy to share. A JSON dictionary with four keys (cells, metadata, nbformat and nbformat_minor) stores all the notebook data.

The cell key holds the code written in the notebook. A notebook document, including cells with code and rich text (i.e., plot, link and documentation), can be executed on a local desktop or a remote server (JupyterLab). The code is executed by a computational engine, the kernel. Multi-language programming is supported by choosing the appropriate kernel. The application comes with the default Python kernel, ipykernel, and it can interface with R by adding the R kernel, irkernel. A file manager, the Notebook Dashboard, can show local files, open new notebooks or shut down kernels. Results can be exported as a Markdown, PDF or HTML file, which can be inspected via a web browser or shared on GitHub.

Both Jupyter and RStudio currently support both Python and R, both for code highlighting and the running environment.

#### 6.1.3. RKward

Launched in 2002, RKWard was one of the first R-specific editors [59], written in C++ for the KDE (https://kde.org/, accessed on 21 April 2022) Linux environment in 2002. Currently, it can run on Windows, macOSX, and Linux operating systems. The project was conceived by Thomas Friedrichsmeier to fulfill the needs of both proficient and neophyte R users. They can leverage the RKWard task-oriented GUI dialogues, including a spreadsheet-like data editor, workspace browser, code editor, help pages and plot preview. R package management is available to easily handle and manipulate R packages. Additional plugins can enhance RKWard’s features. This software may be a great choice for users only working in R.

#### 6.1.4. StatET (Eclipse Plugin)

StatET is an Eclipse-based IDE for both R and Java languages, designed by Stephen Wahlbrink in 2007 [60]. It integrates a number of features for R coding and package building, including an R console, graphic device, editors and R help system. It runs locally or in remote installations of R, and currently supports both Rmarkdown and knitr. Taking advantage of the features of the Eclipse IDE (originally developed in 2001 by IBM for Java), such as the debugging interface and monitoring of environmental variables, StatET is very powerful, and its plug-in system makes it highly customizable [61]. It is preferred by developers already familiar with Eclipse and using it to code in multiple languages. However, Eclipse StatET is not as easy to install and run as RStudio.

#### 6.1.5. Google Colab

A free version of Jupyter Notebook is also available through Google Colab, a cloud-based system not requiring any local installation, and running entirely on Google servers [62]. Colab was developed in 2017 to leverage shared Google CPUs, GPUs and tensor processing units (TPUs), but it provides, at least in the free version, limited space (77 GB) and time (12 h absolute timeout for a single session). Data cannot be permanently stored in the Colab Virtual Machine, but users can import data from local drives, external resources or by accessing the cloud storage system Google Drive. Live editing is not allowed but a copy of notebooks can be shared as a Drive file. Running R in Colab is possible by using rpy2 package in the Python runtime, or by starting a new notebook in the R runtime (https://colab.to/r, accessed on 21 April 2022). Ultimately, Google Colab is a game-changing concept for outsourcing calculations, with an immense potential for the future of R, bioinformatics and data science in general, but we believe it is currently not fully suited for R-dedicated projects.

#### 6.1.6. Visual Studio Code

Visual Studio (VS) Code is a source-code language-agnostic editor, developed by Microsoft for all major platforms. The first free and open source version was published in 2015. Like RStudio and Jupyter, VS Code provides intelligent code completion, debugging, plotting, remote execution, subversion, and much more. Its extension-based architecture allows users to expand its functionalities and add new languages to the environment, including R [63]. The R Extension for Visual Studio Code by Yuki Ueda is a plugins for the VS Code IDE, providing programming functionalities specific to R [64]. The availability of such R-specific add-ons is likely to attract an increasing number of expert programmers to the R community, since VS Code is currently considered one of the most popular developer environment tools in the world, at least according to the Stack Overflow 2021 Developer Survey [65].

### 6.2. Text Editors

In some instances, IDE and GUI features are not required and users may need a lightweight and simpler environment to access local or remote files. In this section, we present a selection of popular text editors, with extension capabilities suited for R programmers.

#### 6.2.1. Vi/Vim

Vi is the visual version of the ‘ex’ command-line editor. It was originally written by Bill Joy, a graduate student at Berkeley in 1976, and officially published with the first BSD Unix release, in March 1978. About 24% of responders to Stack Overflow’s 2021 developer survey asserted they used vi regularly [65]. This editor operates in “Visual” or command mode, which renders the text being edited in a terminal, and “Insert” mode, where text is included in the document, by typing “I”. A “Vi Improved” version, Vim, was released in 1991, and included more features, such as syntax highlighting, mouse support, and new editing commands. Since 2000, Vim has been included with almost every Linux distribution, and GUI versions exist for other operating systems, such as gVim for Linux, and MacVim for macOS. Many plugins improve the Vim user experience for R programmers, such as Nvim-R-Vim [66] for integrating Vim with the R console.

#### 6.2.2. Emacs ESS

Emacs (Editor MACroS) is a family of text editors written in 1976 by David A. Moon and Guy L. Steele Jr. at MIT AI Lab, for the line editor known as Tape Editor and Corrector (TECO [67]). It is one of the two main contenders (the other being Vi) in the traditional editor war [68]. For a new user, both Vi and Emacs are nearly equivalent when it comes to performance, customization and learning effort [69]. The Emacs Speaks Statistics (ESS) package was developed in 1997 by a team led by Antony Rossini and including some R Core Team members, to guarantee a good integration between Emacs and R [70]. ESS is not formally an IDE, but offers functionalities similar to those of RStudio, such as code completion, interactive help and debugging tools.

#### 6.2.3. Sublime Text

An alternative option to the long-standing “Vi or Emacs” question, Sublime Text stands as a popular and general text editor with an attractive and simple user interface. First conceived as an extension for Vim by Jon Skinner in 2008, Sublime Text is a cross-platform, multi-language and highly customizable text editor. The full version is available under a one-time license, but the unregistered version is available for user evaluation without a time limit to purchase a license. Sublime Text can be integrated with R by installing specific extensions, such as R-IDE (https://github.com/REditorSupport/sublime-ide-r, accessed on 21 April 2022), plus common extensions to run the scripts directly into R, such as SendCode (https://github.com/randy3k/SendCode, accessed on 21 April 2022).

#### 6.2.4. Notepad++

Another simple and lightweight code-agnostic text editor is Notepad++, a completely free alternative to Sublime Text for editing R scripts. It was released by Don Ho in 2003 and is only available in Windows environments (analogous editors in MacOSX would be TextWrangler or BBedit). The Notepad++ editor features prepackaged extensions supporting about 80 programming languages; amongst these, the NppToR plugin (https://github.com/halpo/NppToR, accessed on 21 April 2022) provides R language syntax highlighting, folding and auto-completion, on top of the possibility to communicate directly with the R Windows console.

## 7. R and Other Programming Languages

There was no “best” language for bioinformatics and data science at the time of writing. Several studies have attempted to systematically compare programming languages for bioinformatics, in terms of speed of execution of common tasks, input/output performance for large files and more subjective parameters such as lines of code needed for common tasks [71]. Ultimately, the success of a language in fields as large and complex as bioinformatics and data science is the combination of language design, popularity, and availability of libraries, pragmatic solutions, and a community to help with performing specific programming tasks.

R, as we have seen, was historically developed at the intersection of existing languages, such as C, S and Scheme, and has always allowed the usage of native C and Fortran code to perform certain calculations. One example is the *Rcpp* package, which constitutes the main interface to link R with C and C++ code [72]. The synergy of R with other languages is due to the intrinsic energy and open-mindedness of the R community, which is always ready to compare R solutions with those provided by other languages, and is constantly up to speed with the most recent statistical and data technologies; for example, it provided the first tools to perform microarray and RNA sequencing analysis [73,74]. In terms of popularity, R is currently (December 2021) ranked as the 8th most useful language in terms of number of jobs requiring it [75] and as the 7th most popular language according to the PYPL index [9] (Table 2).

Among the top languages shown in Table 2, the three most commonly used for bioinformatics and data science tasks are undoubtedly Python, R and MATLAB.

MATLAB, although computationally and algorithmically efficient, is not freely available and, as such, is more difficult to access by the wider public. On the other hand, Python and R are both freely accessible languages, with Python being the most popular language in the world due to its versatility beyond data analysis (e.g., in database management and website development). Historically, Python and R have grown at the expense of the previous “king” of bioinformatics languages, PERL, which has gradually disappeared since 2005 in Google searches and, in general, from the bioinformatics and data science world (Figure 3C).

R maintains, and probably will maintain, a niche but dominating presence in the field of bioinformatics, providing ready-made packages and functions, alongside a vibrant and helpful community, for most common pipelines in computational biology, from pathway analysis [76] to the conversion of gene identifiers [77].

Performance-wise, R has recently achieved excellent benchmarks in the speed of common matrix operations, when compared to Python [78]. This is shown by the functionality provided by the *data.table* and *dplyr* packages, and in general by the more recent *tidyverse* collection.

## 8. Machine Learning and Artificial Intelligence through R

R has historically been generated by some of the most active statisticians in the world, intertwining its destiny with the development of statistics itself and its implementations, including the most recent explosion of machine learning and artificial intelligence (AI). Indeed, some of the fathers of machine learning, such as Trevor Hastie and Robert Tibshirani, are R programmers and remain active R library developers [79]. One of the allegedly most successful methods to perform machine learning, LASSO, has been available in R virtually since its birth [80], allowing users to readily apply LASSO regression and feature reduction to bioinformatics [81]. 

Several hundreds of R libraries performing a number of machine-learning tasks and methods exist on CRAN. The package *caret* (Classification and REgression Training), written by Max Kuhn, provides a common interface for many of these methods, acting as an umbrella system to run most of machine learning tasks in R with common functions (https://topepo.github.io/caret/, accessed on 21 April 2022). More specifically, *caret* provides functions to perform data splitting and preprocessing, feature selection, model tuning and training, variable importance estimation and model testing. Virtually all machine learning algorithms written for R can still be used as standalone packages or via *caret*, which imports them as needed. *caret* divides methods into regression methods (to predict continuous numerical variables) and classification methods (to predict categorical variables), and continuously adds new methods as they are created (for a full list of the models supported by *caret*, see the CRAN page at http://topepo.github.io/caret/available-models.html, accessed on 21 April 2022). The following code snippet shows how the model training is performed via the R *caret* package. The object “input” contains an input training data frame with variables as columns and observations as rows. The vectors “predictors” and “outcome” contain the variables to be used as predictors and to be predicted, respectively. The object “trainmethod” specifies how to perform the training (e.g., by leave-one-out or by cross-validation). The “metric” parameter specifies the type of output (in this classification case the values to generate a Receiver Operating Characteristic curve, ROC) Finally, the variable “mymethod” can be changed according to the user’s choice, and in this case is set to “gbm”, a gradient boosting model method very popular in regression and classification analyses [82,83]. 



library(caret)





mymethod<-"gbm"





model<-train(data.matrix(input[,predictors]),trainDF[,outcome], 





       method=mymethod, 





       trControl=trainmethod, 





       metric="ROC",





)



More recently, Max Kuhn and RStudio developed a *caret* version running under the tidyverse called *tidymodels*, which covers most of caret functionality with classic tidyverse syntax [84].

Beyond well implemented and readily usable methods, a language for machine learning needs an efficient system for dealing with large datasets. Since at least version 2.0.0, R has been designed to allow the treatment of big data, which are the fuel of artificial intelligence besides the algorithms themselves. This was undertaken via the implementation of lazy loading (see before) to allow datasets to be accessed without being fully loaded in memory, in addition to other methods to operate on sparse data in a highly optimized framework, such as the *data.table* package [26]. R can also recover from insufficient memory errors by notifying the user of the infamous “Cannot allocate a vector of size x” error; such errors do not shut down the R console, allowing the user to recover the current session.

Modern machine learning systems and libraries for applications such as computer vision and language understanding have gained significant popularity among data scientists, and have virtually all been ported to R. For example, Keras, a high-level neural network library, is currently available for R users through at least two different packages: the *kerasR* package, authored and created by Taylor Arnold, and RStudio’s *keras* package. Both packages provide an R interface to the native Python deep learning code, providing a flexible interface for specifying machine learning models. Another popular Python-based machine learning library, Scikit-learn, can be used on R via the *reticulate* package, which provides a comprehensive set of tools for interoperability between Python and R, including the possibility to convert R objects into Python objects and vice versa.

Beyond *caret*, other method-aggregating packages are currently available to run machine learning code in R. The *h2o* package, for example, written by Erin DeLell, provides an interface for the very popular H_2_O machine learning platform, including a plethora of methods from generalized linear models to deep neural networks, and a very convenient automated algorithm called H_2_O AutoML [85]. Another popular package to perform various machine learning tasks is *mlr3*, written by Michel Lang; *mlr3* has been recently rewritten in a fully object-oriented fashion, taking advantage of both *data.table* and the new lightweight R6 classes [86].

A few other R packages available to perform dedicated machine learning tasks are also worth mentioning. The library *MASS*, for example, originally developed for the S language, is still widely used to perform statistical learning tasks such as linear discriminant analysis and quadratic discriminant analysis [87]. Another package is *prophet,* written by Sean Taylor, whose name aptly describes its main task: the prediction of future trends based on time series data [88]; the *prophet* algorithm is elegantly written to account for missing data, outliers and typical phenomena of time series, such as seasonal trends and holiday effects, and is broadly used in various data science studies [89,90].

## 9. R on the World Wide Web: The Shiny Framework

Data is the backbone of research, and its collection and exchange currently constitute one of the central activities of the World Wide Web (WWW). The WWW, since its official birth in 1989 [91], has always been meant as an information-sharing device between scientists, reducing geographical and political barriers. Through the years, the WWW has evolved from a collection of static hypertext documents, passively browsed by users (the Web 1.0 era, also referred to as the Web of Cognition), to a human-centered communication system (the Web 2.0 era, also known as the participative or social web), finally becoming a networked digital technology supporting human cooperation (the Web 3.0 era, or the semantic web) [92]. It has been estimated that about 2.5 quintillion bytes of data are created each day [93], and 4.66 billion active Internet single users were mapped worldwide on January 2021 [94].

In the field of biology and bioinformatics, in parallel with the development of the WWW, the advent of microarrays in the 1990s and of Next Generation Sequencing in the 2000s has generated large quantities of data, collecting quantitative information on “omes” (genomes, transcriptomes, proteomes, metabolomes, interactomes), on drug–target interactions, on protein structures, and much more [95]. Bioinformatics repositories to store this information have been created on the WWW, such as the NCBI’s Gene Expression Omnibus [96] and EMBL-EBI’s ArrayExpress [97] for omics data, Protein Data Bank for protein structures [98], Biogrid and STRING for interactions [99,100], etc. Although access to data has become very easy, data handling and analysis need to be accomplished by well-trained experts (bioinformaticians, computational biologists, data analysists, etc.), usually requiring advanced computational skills and the knowledge of at least a programming language [101].

To enable researchers lacking these requisites to easily perform their own analysis on omics data, many free web applications have been developed in the last few years, which frequently rely on a newly born R system for generating dynamic web sites: the Shiny R framework, implemented in the homonymous *shiny* package [102]. Since R is one of the most popular languages in the world, specifically amongst data scientists, bioinformaticians, and teachers [103], the development of biology-oriented web sites written through Shiny R has flourished, and contributed the highest number of web tools amongst all programming languages; for example, during the COVID-19 pandemics [30], providing tools for mutation tracking [104] and drug–target prediction [105].

In its simplest explanation, Shiny is a framework for building interactive and dynamic web sites taking advantage of the plethora of R functionalities. After its release in 2012, Shiny has continuously grown in popularity, and the general impact of Shiny in peer-reviewed academic literature has been recently discussed [106]. Shiny is well integrated with RStudio, which can readily generate a backbone Shiny website from a template, allowing simple websites to be set up in a matter of minutes (a tutorial is available at https://shiny.rstudio.com/tutorial/, accessed on 21 April 2022).

Technically, a Shiny app is essentially built using two key components, frequently contained in a single app file (*app.R*), or explicitly separated into two files: a user interface (UI) object (usually saved as *ui.R*), and a server function (*server.R*). The UI contains the instructions defining the appearance of the app, enabling the user to interact with the app by clicking on interactive buttons, text boxes or drop-down menus. Default choices are coded into the UI. The server function defines how the app should work, by housing all the instructions that drive the functionality of the application and accessing all built-in functionalities available to R users. The simplest way to create a Shiny app is to create a new directory hosting both ui.R and server.R files. Once running, these files will be used to tell Shiny how the app should both look and behave, respectively. Shiny takes advantage of reactive programming, a style of programming that creates software that responds to events rather than solicits inputs from users [107]. The reactive programming paradigm relies on streams of time-ordered sequences of related event messages allowing objects to be updated in response to changes introduced by the users. Although the R language makes use of imperative programming, where each command should be re-issued each time to make changes in the output, the output is almost instantly updated in Shiny web apps because of reactive programming, where changes to the output value take place whenever each value or object that is connected to the output changes. To create a reactive context, reactive expressions describing how inputs and outputs are connected (the reactive graph) need to be declared in the app, creating object classes (called reactive values) by using the reactive() function. Two objects are required: a source (usually, a user input that occurs through the web interface), and an endpoint (an output object, for example in the form of a table or a plot). It is also possible to place reactive components to manage complex operations, called reactive conductors, between the sources and endpoints. A detailed description of reactivity in Shiny can be found at https://shiny.rstudio.com/articles/reactivity-overview.html.

Hosting of Shiny apps can be performed through a Shiny server (https://www.rstudio.com/products/shiny/shiny-server/, accessed on 21 April 2022) on a private server or on cloud-based hosting services such as Amazon Web Services or Microsoft Azure. Limitations exist, however, for the free version of Shiny server, which, for example, limits the number of usable threads (and therefore of concurrent users) for the deployed Shiny app. The relative simplicity of Shiny, at least for simple web interfaces, and the vast availability of Shiny tutorials and solutions online, makes it an ideal, and currently very successful [102], instrument for bioinformatics to release their methods to all biologists, taking advantage of R code and implementations that would be otherwise inaccessible to non-programmers.

## 10. User-Friendly Resources for Learning R

When learning a new topic, and specifically a programming language, a structured book is never a bad choice. Nonetheless, there are several resources online beyond canonical books, in the form of blogs, mailing lists, forums and general discussion areas. The following paragraphs provide a non-exhaustive list of some of the most useful and popular resources to approach this language, both from a beginner point of view and for more veteran coders.

### 10.1. Books

“The R book” by Michael J. Crawley (ISBN-13: 978-0470973929). An excellent start written by a true R enthusiast, showing with practical examples the basics of R structures and functions, and venturing into early machine learning.

“R for dummies” by Andrie de Vries and Joris Meys (ISBN-13: 978-1119055808). From the popular Wiley series, this book explains R in detail, even for a complete beginner of computer science, bringing him or her up to speed with the most recent R functionalities.

“R for Data Science” by Hadley Wickham (ISBN-13: 978-1491910399). This book, written by the creator of *ggplot2* and *tidyverse*, follows the classic O’Reilly teaching tradition, providing not only a structured lesson, but also a handful cookbook of solutions for the easier and less easy data science operations with R.

“The R inferno” by Patrick Burns (ISBN-13: 978-1471046520). This is one of the least statistics-oriented books, focusing rather on subtleties of the R language and teaching optimal ways to write efficient (in terms of computational time and memory usage) R code.

“Use R!” by Robert Gentleman, Kurt Hornik and Giovanni Parmigiani (ISSN: 2197-5736) is a book series of almost 100 volumes dedicated to practical and focused R usages, written by some of the original founders of the R community itself.

“The R Series” (https://www.routledge.com/Chapman--HallCRC-The-R-Series/book-series/CRCTHERSER, accessed on 21 April 2022) is another collection of R books edited by John Chambers, Torsten Hothorn, Duncan Temple Lank and Hadley Wickham. This series consists of 62 series titles, and covers a wide range of topics, from more hardcore programming concepts to applications of R in biology, finance and criminology).

### 10.2. Online Resources

“R-bloggers” (www.r-bloggers.com, accessed on 21 April 2022) is a community effort to provide the latest news on R and the R world, describing new packages and analyzing the current trends and solutions for data science and bioinformatics.

“One R tip a day” is a prolific Twitter account (@rlangtip) providing extremely smart and high-level notions on R. Although not suggested as a starting point, the tweets contained here can be extremely useful for the expert R programmer in need of more tricks in their toolset. An analogous Twitter account, @Rfunction_a_day, provides similar daily information on more complex useful functions.

“Coursera R Programming”, by Roger Peng (https://coursera.org/learn/r-programming, accessed on 21 April 2022). Coursera.org, in addition to other similar websites, such as Datacamp.com, provides online classes to learn R and several R-specific topics, such as bioinformatics and data science. Although not free, these courses are popular entry points for users that favor a human teacher over reading, or prefer a mixture of both.

“The R-Podcast” (https://r-podcast.org/, accessed on 21 April 2022), is a YouTube channel depicting 1 h long focused topics on the R world, from the basics to more complex topics. Hosted by Eric Nantz, this channel often features interviews with the authors of R packages and solutions, such as shiny and tidymodels, aiming also at giving a face and a voice to the contemporary R community.

“Posts you might have missed” (https://postsyoumighthavemissed.com/, accessed on 21 April 2022) by Alastair Rushworth is a monthly digest of news from the R (and Python) world, focusing on data science. Very useful as an aggregator of new guides, packages, books and developments for R programmers. A daily version by the same author is available on the Twitter account @icymi_r.

“The R documentation” (https://www.r-project.org/other-docs.html, accessed on 21 April 2022) is a web site maintained by the R Foundation itself, linking to a vast quantity of books and learning material on R, both general and specific. Material is available in English, Chinese and Russian.

## Figures and Tables

**Figure 1 life-12-00648-f001:**
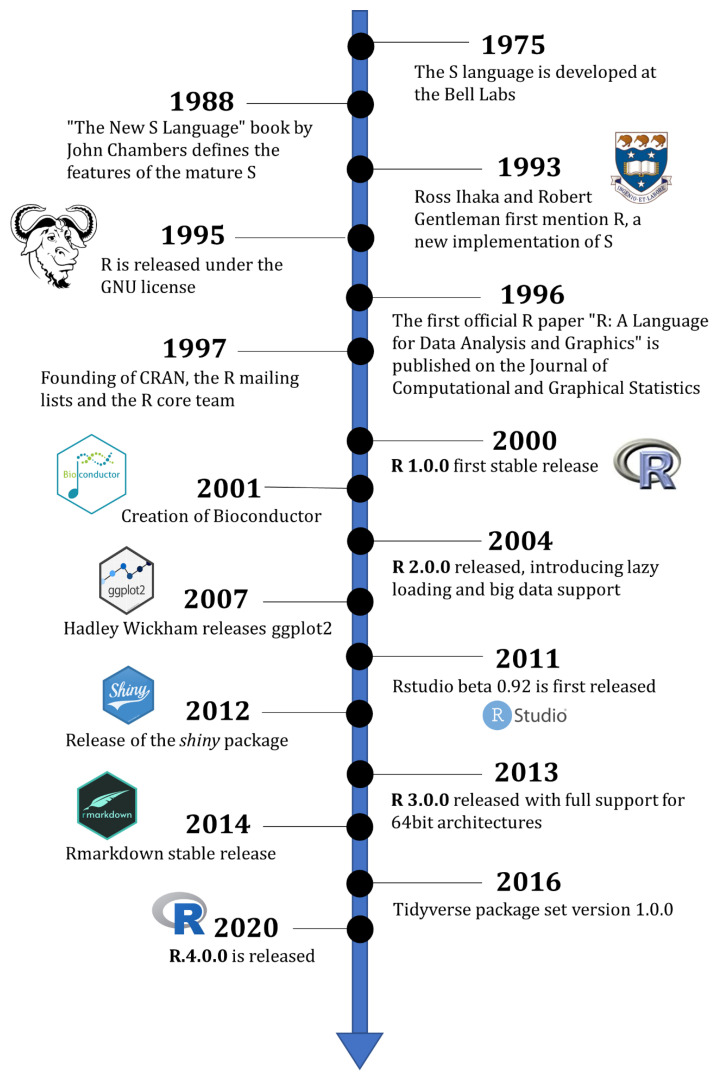
Timeline of R history with selected milestones.

**Figure 2 life-12-00648-f002:**
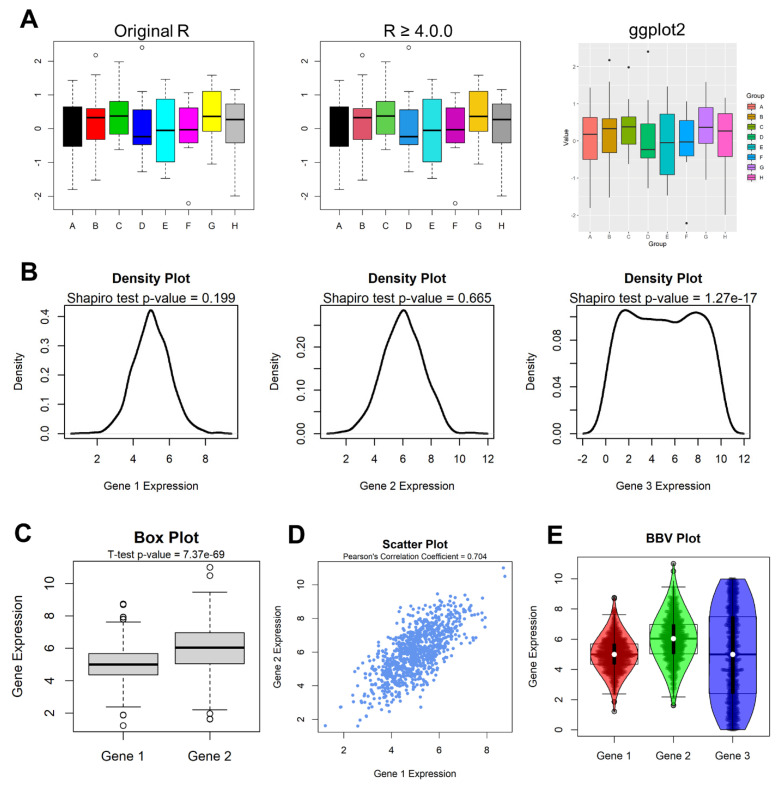
(**A**) Box plots drawn using the default R boxplot() function in original R (**left**), R since 4.0.0 (**middle**) and ggplot2 (**right**). (**B**) Density distribution plots for three distributions, combined with the results of the Shapiro–Wilk test. (**C**) Default R boxplot comparing two distributions and providing the output *p*-value of the Student’s *t*-test. (**D**) Scatter plot indicating the co-expression of two genes, and the Pearson’s correlation coefficient of the joint distribution. (**E**) Example of overlapping different plot types in R: box plot, beeswarm plot and violin plot (BBV Plot) for three numeric distributions (called Gene 1, Gene 2 and Gene 3).

**Figure 3 life-12-00648-f003:**
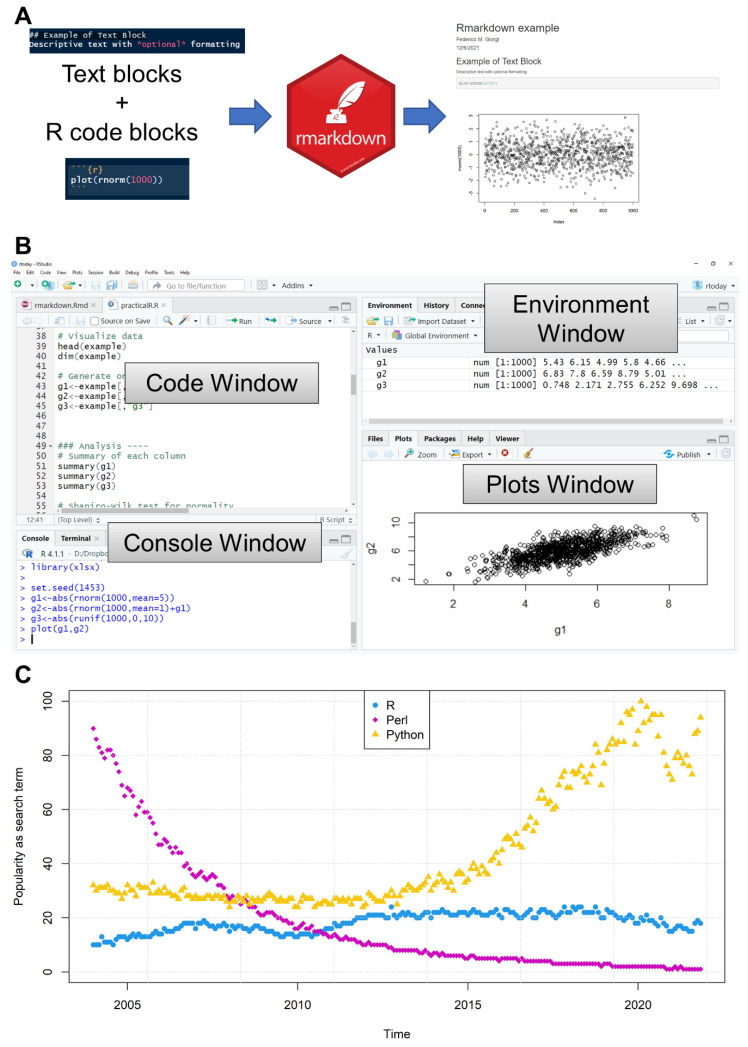
(**A**) Concept diagram of how Rmarkdown can merge text and code blocks to create documents. (**B**) Example of an R IDE, RStudio, showing multiple elements to assist the R programmer. (**C**) Worldwide popularity of search terms “R”, “Python” and “Perl” in the years 2004–2021 (source: Google Trends). The topic of the search term is limited to “Programming Language”.

**Table 1 life-12-00648-t001:** Selected list of text editors and IDEs for R programmers.

Text Editor/IDE	Release Year	Web Link
RStudio	2011	https://www.rstudio.com
Jupyter Notebook	2014	https://jupyter.org
RKWard	2002	https://rkward.kde.org
Eclipse StatET	2010	https://projects.eclipse.org/projects/science.statet
Google Colab	2017	https://colab.research.google.com
Visual Studio Code	2015	https://code.visualstudio.com
vi/Vim	1976	https://www.vim.org/download.php
Emacs ESS	1997	https://ess.r-project.org/
Sublime Text	2008	https://www.sublimetext.com/
Notepad++	2003	https://notepad-plus-plus.org/downloads/

**Table 2 life-12-00648-t002:** Most popular programming languages according to the 2021 PYPL index.

Rank	Language	Share	Trend
1	Python	30.21%	−0.50%
2	Java	17.82%	1.30%
3	JavaScript	9.16%	0.60%
4	C#	7.53%	1.00%
5	C/C++	6.82%	0.60%
6	PHP	5.84%	−0.20%
7	R	3.81%	0.00%
8	Swift	2.03%	−0.20%
9	Objective-C	2.02%	−1.60%
10	MATLAB	1.73%	−0.10%

## Data Availability

Not applicable.

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
