# Peer review of "The R Language: An Engine for Bioinformatics and Data Science"

_life, 2022, doi:10.3390/life12050648_

Round 1

Reviewer 1 Report

The authors provide an almost complete overview of the features and capabilities of R. It is therefore a good reference for anyone considering working with R, and for every R user who wants to learn more advanced features. I have only a few minor comments:

p.4, 4th line: Professors -> Professor

p.5, paragraph 2.5: … it soon became apparent that a specialized project for Bioinformatics R users. This part of the sentence is incomplete. Please add a verb, e.g. “was needed”, at the end of the sentence.

p.8, paragraph 3.1: The CRAN repository currently hosts 19,027 packages. Please add the date when you accessed the CRAN website.

p.9, paragraph 3.2: Bioconductor is the second-largest R package repository, hosting 3,422 packages. Please add the date when you accessed the Bioconductor website.

p.20, paragraph 8: Apart from caret, another package includes many machine learning algorithms: mlr, (see https://cran.r-project.org/web/packages/mlr/vignettes/mlr.html ). I suggest also discussing the mlr package and compare with caret.

Reviewer 2 Report

The manuscript is effectively a broad presentation of R. It is probably a useful document. It is not clear to me why the document should be published as a peer-reviewed article given that it is not a scientific document. However, there is probably little harm in doing so.

The document is generally fine and readable, but the following should be corrected in the final version:

  1. Paragraphs that include a single sentence, especially when this sentence is superfluous, should be avoided. E.g., see the paragraph at the end of section 2.
  2. In section 2.7, you have an hyperlink in the final paranthesis of an URL.
  3. Why an upper case in the word "bioinformatics"?
  4. Use unbreakable spaces where needed (e.g., last line of section 2.2).
  5. The authors should moderate their use of adverbs such as 'very' which are abused in the main text.
  6. Once you have introduced an acronym, such as IDE, please do not spell it out again and again.
  7. Avoid having sections such as 6.1 that begin immediately with a subsubsection. It indicates a poor structure.

I am not an R expert per se, thus it is possible that some of the statements are incorrect. I would not necessary catch such factual errors.

Given that there is no scientific research, there is little else for me to comment on. I leave it up to the editor to decide whether this manuscript should be published.

Reviewer 3 Report

The authors reviewed the history of the development of the R Language and focused on bioinformatics. The manuscript is well written and could serve as a primary reference for students get in bioinformatics or computational biology. I do not find major concerns.

Minor concerns:

In the 2.6 Expanding R: ggplot2 and the tidyverse section. Please change Ggplot2 to ggplot2, I think that the second is the formal name for the library.

In the overall text, sometimes the name of libraries are italicized and sometimes not, please italicize all library names to make all the text uniform.
